# Critical properties of quantum three- and four-state Potts models with boundaries polarized along the transverse field

**Natalia Chepiga**

Kavli Institute of Nanoscience, Delft University of Technology,
Lorentzweg 1, 2628 CJ Delft, the Netherlands

n.chepiga@tudelft.nl

## Abstract

By computing the low-lying energy excitation spectra with the density matrix renormalization group algorithm we show that boundaries polarized in the direction of the transverse field lead to scale-invariant conformal towers of states at the critical point of the quantum four-state Potts model - a special symmetric case of the Ashkin-Teller model. Furthermore, by direct comparison of the excitation spectra we phenomenologically establish the duality between the transverse-polarized and three-state-mixed boundary conditions at the four-state Potts critical point. Finally, for completeness, we verify that in the quantum three-state Potts model the "new" boundary conditions dual to the mixed ones can be realized by polarizing edge spins along the transverse field.

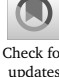

# 1 Introduction

Over the past decades boundary critical phenomena attracted a lot of interest in the context of statistical physics [1–3] and impurity problem [4–6] in condensed matter and particle physics. The presence of the boundary affects measurable observables and change energy spectra making the problem highly non-trivial [7, 8]. Many exact results for the simplest critical models and the simplest sets of boundary conditions have been predicted by the boundary conformal field theory [1, 2, 7, 9–11].

The attention to the boundary critical phenomena has been re-attracted recently by the progress in numerical techniques for quantum many-body systems. Over the years, density matrix renormalization group (DMRG) algorithm [12–15] has established itself as one of the most powerful and accurate numerical tool for low-dimensional systems. Although DMRG is suitable for systems with either open or periodic boundary conditions, the latter has significantly higher computational costs. Thus, numerical investigation of the nature of quantum phase transitions often requires a theoretical understanding of the boundary critical phenomena. Traditionally, the universality class of the transition is identified numerically by computing critical exponents and the central charge. Both can be extremely sensitive to finite-size effects and affected by logarithmic corrections or possible crossovers. Excitation spectra at the conformal critical point are known to form a special structure - conformal towers of states - and contain more information about the underlying critical theory. Therefore, the catalog of the conformal towers of states for different critical theories and under various boundary conditions is essential for numerical investigation of quantum phase transitions.

For the critical transverse-field Ising model the exact correspondence between primary fields and various combinations of free and fixed boundary conditions has been worked out analytically by Cardy [2] and further confirmed numerically [16–19]. For the tri-critical Ising model Affleck [11] predicted partially-polarized boundary conditions to be conformally invariant and different from the free and fully-polarized ones. This prediction has been recently confirmed numerically in 1+1D [20] and 2+0D [19].

The three- and four-state Potts models are generalization of the transverse-filed Ising model to a system with local Hilbert space $d = 3$ and 4 respectively and can be defined by the Hamiltonian [21]:

$$H_{\text{Potts}} = -J \sum_{i=1}^{N-1} \sum_{\mu=1}^{d} P_i^\mu P_{i+1}^\mu - h \sum_{i=1}^{N} P_i \,, \tag{1}$$

where $P_i^\mu = |\mu\rangle_{ii}\langle\mu| - 1/d$ tends to project the spin at site $i$ along the $\mu$ direction while $P_i = |\eta_0\rangle_{ii}\langle\eta_0| - 1/d$ tends to align spins along the direction $|\eta_0\rangle_i = \sum_\mu |\mu\rangle \sqrt{d}$. The first term in the Hamiltonian plays the role of the ferromagnetic interaction, while the second one is a generalized transverse field. The model is critical for $h = J$. In appendix A we provide alternative definition of the model used in the literature. For convenience, we label single-particle states for $d = 3$ by A, B and C. The boundary-field correspondence for free, fully-polarized (A, B or C), and mixed [3] (AB, AC or BC) boundary conditions has been established in the original work by Cardy [2]. In general, restricting the local Hilbert space at the boundary to take the values in $\{1, 2, ..., Q_1\}$, that is, in a subset of the original range $\{1, 2, ..., Q\}$ of the ferromagnetic $Q$-state Potts model are also known as blob boundary conditions. Blob boundary conditions naturally include free boundary conditions when $Q_1 = Q$ and fixed boundary conditions when $Q_1 = 1$. For the ferromagnetic three- and four-state Potts models blob boundary conditions are conformally invariant [22, 23].

Later, Affleck, Oshikawa and Saleur [24] have found that the fully-polarized boundary conditions are dual to the free ones and predicted the "new" conformally-invariant boundary conditions dual to the mixed ones. This completes the set of the conformally-invariant bound-

ary conditions for the three-state Potts critical point [25]. Conformal tower of states with the "new" boundary conditions has been recently detected numerically in 2+0D by allowing negative entries in the boundary Bolzmann weight matrix [19]. It is therefore not obvious how to discuss the new boundary conditions in terms of the original local Hilbert space and without invoking the duality.

However, quantum 1D chains allows a simpler physical realization of the new boundary conditions. In the original paper [24], the authors got an indication that the new boundary conditions can be stabilized by reverting the sign of the transverse field at the edges. Moreover, it turns out that the new critical point is stable with respect to the magnitude of the boundary transverse field. In other words, the new boundary conditions can be expected when the edges are polarized in the direction of the transverse field. Below we will provide the numerical evidence confirming this field-theory prediction.

The main goal of this paper is to show that the concept of the new boundary conditions can be generalized beyond the three-state Potts model. Up to date, only two main classes of conformally invariant boundary conditions of the four-state Potts and the Ashkin-Teller models have been studied in the literature: various types of closed loops including periodic, anti-periodic and twisted boundary conditions [26, 27], and open chains with the blob boundary conditions such as free, fixed and mixed [9, 28]. In the preset paper we will show that in analogy with three-state Potts model, there is yet another type of "new" conformally invariant boundary conditions that can be realized by polarizing the edge spins in the direction of the transverse field. Relying on extensive numerical simulations we will show that these transverse-polarized boundary conditions are dual to the three-state mixed ones, i.e. those where only one out of four single-particle states is excluded at the edges so $Q_1 = 3$ and $Q = 4$. This provides a motivation and a phenomenological starting point for further development of the boundary conformal field theory beyond the simplest loop and blob boundary conditions in the critical Ashkin-Teller model.

The rest of the paper is organized as follows. In Section 2 we briefly review the numerical method used in the paper. In Section 3 we verify that excitation spectra of the critical three-state Potts model with transverse-polarized boundary conditions correspond to the conformal tower of state with new boundary conditions. Section 4 is dedicated to the boundary critical phenomena in the four-state Potts model. In Section 4.1 we benchmark our method by verifying the duality between free and fixed boundary conditions in the four-state Potts model. In Section 4.2 we present numerically extracted conformal towers of states of the four-state Potts model with transverse-polarized boundary conditions and show their duality the three-state mixed boundary conditions. The results are summarized and put in perspective in Sec.5.

## 2   The method

Our numerical simulations have been performed with an extended version of the DMRG algorithm explained in details in Ref. [16]. In this section we briefly review the main features of the algorithm and provide model-specific technical details.

The DMRG [12] algorithm has been originally designed to search for the ground-state. It provides an efficient low-entanglement approximation of quantum many-body state. The accuracy of the wave-function is controlled by the dimension $D$ of the tensors - the number of basis vectors in the density matrix with the largest Schmidt values. Calculation of the excitation spectra is usually more involved. If the wave-function obeys some symmetry, and if the excited state of interest is the lowest energy state of some symmetry sector, then the energy of this state can be computed by running the ground-state DMRG within the corresponding sector. This is a common practice to compute magnetic excitations in spin chains [12, 29–31]. If,

however, excitations cannot be distinguished by any symmetry, as in the case of the three- and four-state Potts models, the algorithm has to be modified significantly. There are three well established strategies. *i)* The density matrix is constructed not only from basis vectors that appear in the Schmidt decomposition of the ground state but mixed with the basis vectors that appear in the Schmidt decomposition of low-lying excitations [32–36]. In this case the bond dimension and the complexity increase very fast with the number of excitations, thus typically one targets five or fewer excited states [13]. *ii)* After constructing the ground-state in the matrix product state (MPS) representation, one can search for an eigenstate that is orthogonal to the ground state and has the smallest energy [15, 36, 37]. Higher excitations can also be accessed by looking for an eigenstate orthogonal to all previously constructed eigenvectors. By contrast to the first approach, the bond dimension remains small, but the algorithm has to be re-run for each eigenstate. *iii)* The third approach relies on the phenomenological observation that for critical systems an approximate basis constructed for the ground state is also suitable to describe the low-lying excited states [16]. By contrast to the first approach this method remains variational with respect to the ground-state. Well converged excitations appear as a flat modes as a function of DMRG iterations. The method has been benchmarked on the critical Ising and three-state Potts models for which the conformal towers with up to 30 states have been computed [16].

We use infinite-size DMRG with the bond dimension $D = 30$ to produce a guess wave-function and increase the bond dimension up to $D = 67$ in the warm-up sweep. In the following six sweeps we increase the bond dimension linearly with each half-sweep up to its maximal value $D_{\mathrm{max}} = 250$. This way we can easily track the convergence with respect to both the number of sweeps and the bond dimension $D$. When convergence cannot be reached for the chosen $D_{\mathrm{max}}$, typically this only happens for large system size and higher excitation levels, we still can get correct estimate of the spectra by extrapolating the energies. In Fig.1 we present one of the trickiest case - the critical four-state Potts model with free boundary conditions and $N = 100$.

In Fig.1**a** we show raw DMRG data for the energy spectrum as a function of DMRG iterations. A periodic increase of the excitation energies occurs close to the chain boundary and is the result of the reduced Hilbert space by MPS construction. The first excited state is three-fold degenerate (yellow and red symbols are almost completely hidden under the purple ones) and starting from the fourth sweep has a well converged energy reflected in the flat intervals. The results for the lowest-lying excitation with free boundary conditions agree within 0.5% with the corresponding Bethe ansatz calculations [9].

The convergence of higher excitations is often slower. When convergence cannot be reached the results are obtained by extrapolating the value of the energy at the local minima towards infinite number of sweeps (or equivalently towards infinite bond dimension $D$). For extrapolation we use a linear fit of the last five points (black lines).

For the three-state Potts model we include only the converged results, without applying an extrapolation. For the four-state Potts model the extrapolation has been applied for higher energy levels for $N > 50$.

## 3 Transverse-polarized boundary conditions for three-state Potts critical point

Let us first verify the realization of the new boundary conditions predicted by Affleck at al. [24] in three-state Potts model defined by the Hamiltonian 1 with $d = 3$ with transverse-polarized boundary conditions. The critical point $h = J$ is described by the minimal model of conformal field theory with $(p, p') = (6, 5)$ and ten primary fields listed in B [38–40]. We

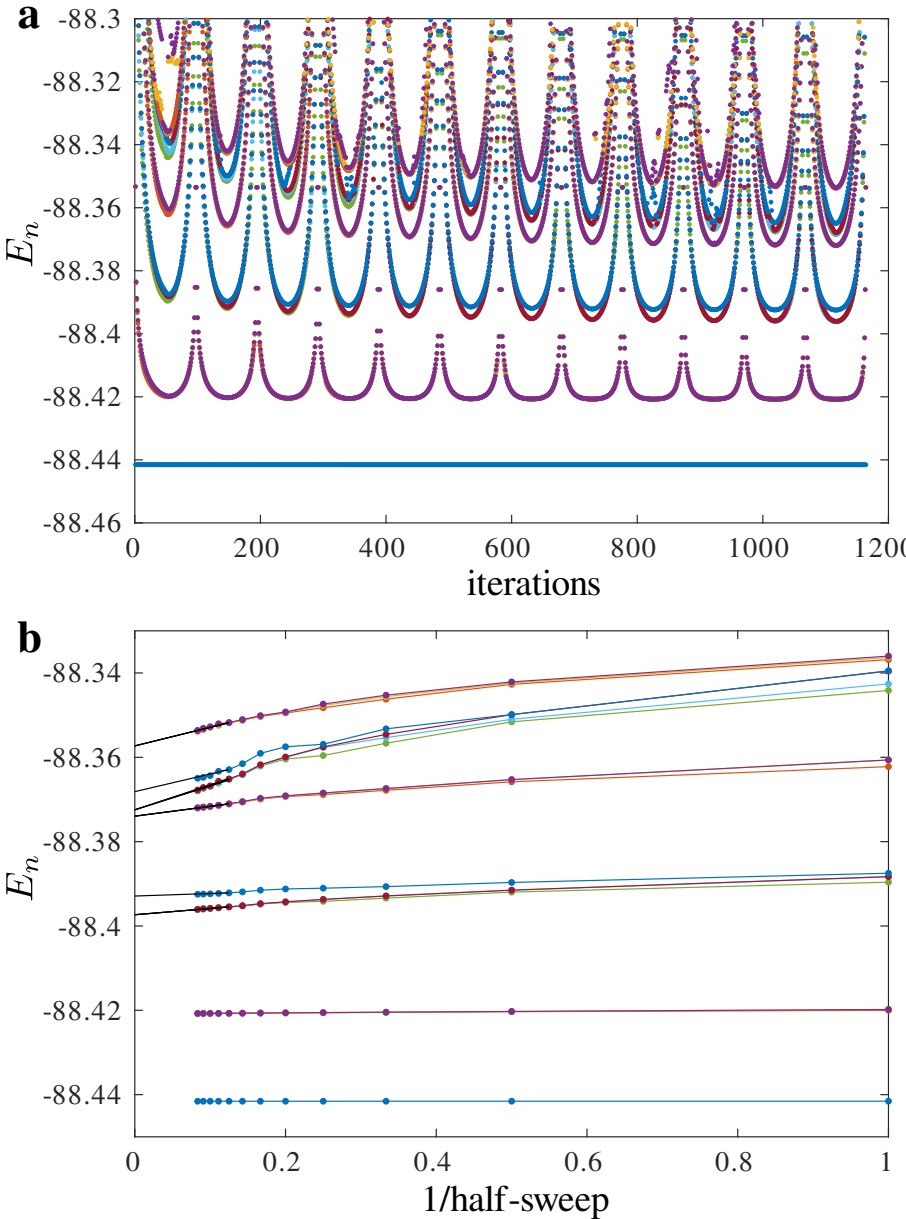

Figure 1: **a** Energy of the 19 low-energy states in the critical four-state Potts model with $N = 100$ sites as a function of iterations. The periodic increase of the energy occurs close to the chain boundary and is the result of the reduced Hilbert space by MPS construction. The flattening of the energies in the middle of the chain is an indicator of convergence. Non converged states are extrapolated towards infinite number of sweeps (equivalently infinite bond dimension) as shown in **b**. Note that many of the shown states are three-fold degenerate and some data points are completely hidden behind the others.

realize fixed boundary conditions A (B, or C) by applying a negative longitudinal field along the first (second, or third) component of the local Hilbert space, while positive longitudinal field along the same component allows to exclude this state and thus lead to mixed boundary conditions BC (AC, AB). In order to check the emergence of the "new" boundary conditions predicted by Affleck at al. [24] we polarize the edges along the direction of the transverse field by setting up the field $h_1 = h_N = -10$ while keeping the transverse field in the bulk critical $h_i = J = 1$ for $2 \leq i \leq N - 1$. Below we remind the list of the partition functions involving the

new boundary conditions [24]:

$$Z_{\text{new,A}} = Z_{\text{new,B}} = Z_{\text{new,C}} = \chi_{2,2} + \chi_{3,2}, \tag{2}$$

$$Z_{\text{new,AB}} = Z_{\text{new,BC}} = Z_{\text{new,AC}} = \chi_{1,2} + \chi_{4,2} + \chi_{2,2} + \chi_{3,2}, \tag{3}$$

$$Z_{\text{new,free}} = \chi_\epsilon + \chi_\sigma + \chi_\sigma^\dagger, \tag{4}$$

$$Z_{\text{new,new}} = \chi_I + \chi_\epsilon + \chi_\sigma + \chi_\sigma^\dagger + \chi_\psi + \chi_\psi^\dagger. \tag{5}$$

The structure of the conformal tower of states are given by the small-$q$ expansion of the corresponding characters listed is the B. The final expression for each set of boundary conditions are provided below:

$$Z_{\text{new,A}} = q^{-1/30+1/40}\left(1 + q^{0.5} + q + q^{1.5} + 2q^2 + 2q^{2.5} + 3q^3 + 3q^{3.5} + 4q^4 + 5q^{4.5} + \dots\right), \tag{6}$$

$$\begin{aligned}Z_{\text{new,AB}} = q^{-1/30+1/40}\bigl(&1 + q^{0.1} + q^{0.5} + q + q^{1.1} + q^{1.5} + q^{1.6} + 2q^2 \\ &+ q^{2.1} + 2q^{2.5} + q^{2.6} + 3q^3 + 2q^{3.1} + 3q^{3.5} + 2q^{3.6} + \dots\bigr),\end{aligned} \tag{7}$$

$$Z_{\text{new,free}} = q^{-1/30+1/15}\left(2 + q^{\frac{1}{3}} + 2q + 2q^{1\frac{1}{3}} + 4q^2 + 2q^{2\frac{2}{3}} + 6q^3 + 4q^{3\frac{1}{3}} + \dots\right), \tag{8}$$

$$\begin{aligned}Z_{\text{new,new}} = q^{-1/30}\bigl(&1 + 2q^{\frac{1}{15}} + q^{\frac{2}{5}} + 2q^{\frac{2}{3}} + 2q^{1\frac{1}{15}} + 2q^{1\frac{2}{5}} + 2q^{1\frac{2}{3}} \\ &+ q^2 + 4q^{2\frac{1}{15}} + 2q^{2\frac{2}{5}} + 4q^{2\frac{2}{3}} + \dots\bigr).\end{aligned} \tag{9}$$

Let us briefly remind how conformal towers can be read-out form the small-$q$ expansion. For example, let us consider the expansion for $Z_{\text{new,free}}$ given by Eq.8. There is a pre-factor that is the same for all towers and defined by the central charge $c$ of the critical theory as $q^{-c/24}$. For the three-state Potts model $c = 4/5$ that results in $q^{-1/30}$. The second pre-factor is the smallest scaling dimension of the primary fields entering the tower. For $Z_{\text{new,free}}$ it is equal to 1/15 - the dimension of the primary field $\sigma$. Since both, $\sigma$ and $\sigma^\dagger$ enter the tower, the multiplicity of the corresponding levels are doubled, in particular, the ground-state is two-fold degenerate which is reflected in the first term in the brackets. All other terms in the expansion are given in the form $mq^n$, where $m$ reflects the multiplicity of the energy level $n$.

In order to extract conformal towers numerically, we compute low-lying energy spectra with up to 21 energy levels. According to the conformal field theory, energy gap scales as $E_n - E_0 = \pi v n/N$, where $N$ is the system size and $v$ is a non-universal sound velocity. For the three-state Potts model defined by the Hamiltonian 1 the value of the velocity is known exactly $v = \sqrt{3}/2 \approx 0.866$ [26] and will be used throughout this section. Our numerical results for the four conformal towers involving transverse-polarized boundary conditions are presented in Fig.2. Given that there are no fitting parameters the agreement between the theory (colored lines) and the numerical data (blue symbols) is extremely good. One can notice that some conformal towers ($\chi_{2,2}$, $\chi_{3,2}$, $\sigma$, $\epsilon$) are affected by finite-size effects stronger than the other ($I$, $\psi$, $\chi_{1,2}$, $\chi_{4,2}$). This effect has been observed before for fixed, free and mixed boundary conditions [16].

To summarize this section, conformal towers of states predicted by Affleck at al. [24] for the new boundary conditions appear in a quantum 1D version of the critical three-state Potts model with boundaries polarized in the direction of the transverse field. This provides a more physical and intuitive realization of the new boundary conditions in quantum chains than in the classical 2D model that requires negative entries in the boundary Bolzmann weight matrix.



Figure 2: Conformal towers of states of the critical three-state Potts model with one edge polarized along the transverse field direction (following Ref. [24] we use the notation 'new') and the second edge is (a) fixed to one of the three single-particle states; (b) mixed between the two single-particle states; (c) free; (d) also polarized along the transverse field. The velocity is fixed to the exact value $v = \sqrt{3}/2$. Blue symbols correspond to our DMRG data (different symbols are chosen to clarify multiplicities), lines of different colors correspond to different primary fields in the tower and listed on the right. The numbers under each character show the expected multiplicities of the levels and always match our numerical data. Blue dotted lines are guide to the eyes indicating where the energy levels are expected to end in the thermodynamic limit and for infinite DMRG bond dimension $D$.

# 4 Boundary critical phenomena in four-state Potts model

Boundary phenomena at the four-state Potts critical point are far less understood. The four-state Potts model defined by the Hamiltonian 1 is a straightforward generalization of the three-state Potts model to four-dimensional local Hilbert space. On the other hand, the four-state Potts critical point is a special symmetric point of a generic Ashkin-Teller critical theory [41]. An effective quantum Ashkin-Teller model can be defined by the following microscopic Hamiltonian:

$$H_{\text{Ashkin-Teller}} = -J \sum_{i=1}^{N-1} \left[ \frac{1+\lambda}{2} \sum_{\mu=1}^{d} P_i^\mu P_{i+1}^\mu - \frac{1-\lambda}{2} (P_i^1 P_{i+1}^4 + P_i^2 P_{i+1}^3 + \text{h.c.}) \right] - h \sum_{i=1}^{N} P_i(\lambda),$$
(10)

where $\lambda$ is the Ashkin-Teller parameter and

$$P_i(\lambda) = \frac{1}{4} \begin{pmatrix} 0 & 1 & 1 & \lambda \\ 1 & 0 & \lambda & 1 \\ 1 & \lambda & 0 & 1 \\ \lambda & 1 & 1 & 0 \end{pmatrix}.$$

The model coincide with the four-state Potts model given by Eq.1 for $\lambda = 1$. In appendix A we provide alternative definitions of the model used in the literature. For $\lambda = 0$ the model is a quantum version of the four-state clock model and corresponds to two decoupled Ising chains. Along the $J = h$ line the model is described by the Ashkin-Teller critical theory with central charge $c = 1$ and critical exponents varying continuously with $\lambda$. The operator content and partition functions on a torus have been analyzed by Yang [42]. The energy spectra of the critical quantum Ashkin-Teller and Potts chains with free boundaries have been obtained by mapping the problem onto the XXZ chain with free boundaries and a complex surface field and solving the latter with the Bethe ansatz [9]. Boundary critical phenomena for the special case of $\lambda = 0$ have been studied recently in the context of a defect line in two-dimensional Ising model [43]. The conformal tower of states as a function of $\lambda \in [0,1]$ for fixed and symmetric boundary conditions A-A has been reported recently in Ref. [44].

There are two challenges associated with numerical investigation of the conformal towers of the Ashkin-Teller model. First, by contrast to the Ising and 3-state Potts minimal models, there are logarithmic corrections present in the Ashkin-Teller critical theory which might significantly affect numerical results [45]. Second, there is an infinite number of primary fields in the generic Ashkin-Teller model. The goal of this section is to demonstrate the duality between the transverse-polarized boundary conditions and the three-state mixed boundary conditions, the special case of the blob boundary conditions with $Q = 4$ and $Q_1 = 3$.

## 4.1 Free and fixed boundary conditions in the four-state Potts model

Our final goal will be to establish the duality between the two sets of boundary conditions and for this we will compare the energy spectra obtained on finite-size clusters. In order to benchmark the method, we start with free and fixed boundary conditions that are known to be dual (see, for instance, Ref. [23]). Following the notations of the previous section we label the single-particle states for $d = 4$ by A, B, C and D. The duality between free and fixed boundary condition implies

$$Z_{\text{Free-Free}} = Z_{\text{A-A}} + Z_{\text{A-B}} + Z_{\text{A-C}} + Z_{\text{A-D}}.$$
(11)

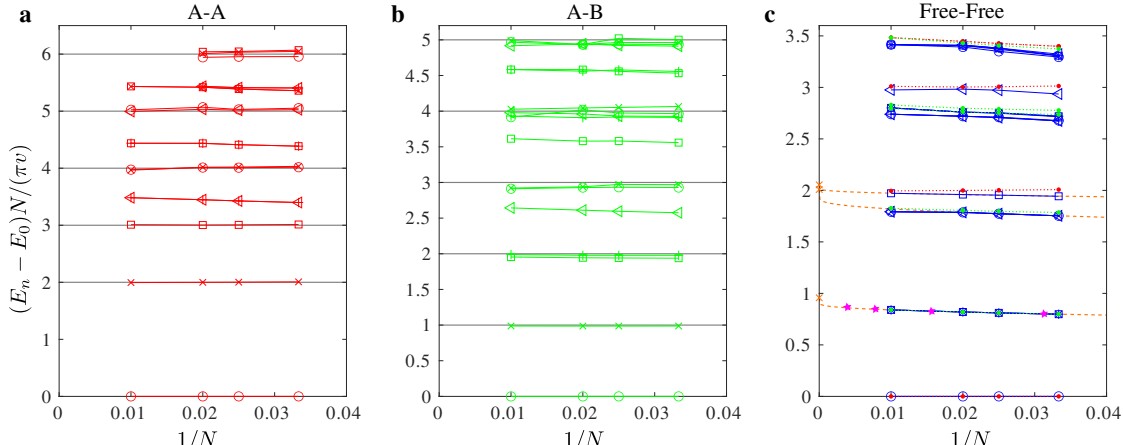

Figure 3: Conformal towers of states of the four-state Potts model with **a** fixed symmetric and **b** fixed antisymmetric and **c** free boundary conditions. Symbols are DMRG data points extracted from the low-lying energy excitation spectra with velocity $v = \pi/4$. The velocity computed from the lowest energy gap in **a** $v \approx 0.785$ is in excellent agreement with this value. Gray lines are integer levels shown for reference. In **c** the results from **a** (red) and **b** (green) are shown as a reference, the latter is shifted such that it starts at the first excites state of the Free-Free tower. Each level in Free-Free tower that matches a level of the A-B tower is three-fold degenerate. Magenta stars are results from the Bethe ansatz calculations [9], the agreement with DMRG data is within 0.5%. Orange dashed lines are finite-size extrapolation assuming log-corrections in the form derived by Cardy [45]. Extrapolated results (orange crosses) agree within 5% with the exact result $x = 1$ [9] and $x = 2$ for higher levels.

Obviously, due to symmetry in the model the first index A can be replaced with either B, C or D; for the same reason the conformal towers of states with A-B, A-C, and A-D boundary conditions are identical. In other words, the energy spectra of a chain with free-free boundary conditions corresponds to superposed energy spectra of a chain with A-A boundary conditions and three-fold degenerate spectra with A-B (or equivalently A-C or A-D) boundary conditions. Let us check this numerically.

In case of symmetric boundary conditions with the same state realized on the left and on the right edges (including A-A and Free-Free) we expect a conformal tower of states to contain the identity tower $I$ with the scaling dimension $x = 0$. The distinct feature of this tower is the absent linear in $q$ term, i.e. the first excited state takes place at the level 2 that corresponds to $q^2$. Precisely this structure we observe in Fig.3(a). We extract the velocity from the lowest energy gap in a chain with A-A boundary condition and get the value $v = \Delta E N/(2\pi) \approx 0.785$ which is in excellent agreement with the exact value $\pi/4$ [26]. We will use this value through the rest of this section.

The identity tower has the smallest possible scaling dimension $x = 0$ implying that the ground-state of a chain with any symmetric boundary conditions (e.g. Free-Free) belongs to the identity tower $I$. In Fig.3(c) we show with blue symbols the excitation spectrum with Free-Free boundary conditions. We compare it with the tower obtained with A-A boundary conditions shown in Fig.3(c) with red dots. The spectrum of a tower with fixed but non-symmetric boundary conditions A-B (equivalently A-C and A-D) is presented in Fig.3(b). Note that in this case we clearly see the excitation at the first level. Since A-A and A-B towers are expected to be the only components of the Free-Free tower we associate the first excited state in Free-Free tower that does not match the A-A one with the ground-state in A-B tower and plot higher levels of the A-B tower with respect to this level. The A-B tower is shown in

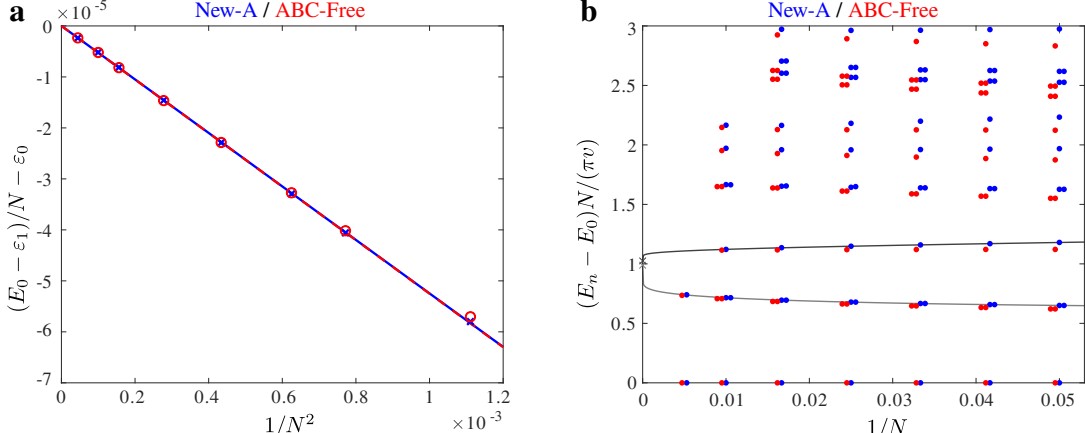

Figure 4: Direct comparison of the conformal tower of states with A-New (blue) and Free-ABC (red) boundary conditions. **a** Finite-size scaling of the universal part of the ground-state energy, where $E_0$ is the total energy of a chain with $N$ sites, $\varepsilon_0$ is an energy per site in the thermodynamic limit, $\varepsilon_1$ is a non-universal contribution from the edges. **b** Conformal towers of states extracted from the excitation spectra with A-New (blue) and Free-ABC (red) boundary conditions. For clarity some data points are shifted horizontally by up to $2 \cdot 10^{-3}$. Black lines are finite-size extrapolation assuming log-corrections in the form derived by Cardy [45]. Up to some minor finite-site effect the two spectra are identical that implies the duality between the corresponding sets of the boundary conditions.

Fig.3(c) in green. Note that each level in Free-Free tower that matches a level of the A-B tower is three-fold degenerate in a complete agreement with Eq.11.

## 4.2 Transverse-polarized and three-state-mixed boundary conditions

Let us now consider the boundary conditions where one state, say D, is suppressed at the edge, which leads to the three-state-mixed boundary condition ABC. As any blob boundary conditions including free and fixed ones, the three-state mixed boundary conditions are conformally invariant [22].

In Fig.4 we present a direct comparison of the energy spectra with ABC-Free and New-A boundary conditions, where following the notation by Ref. [24] we use the name "New" for the edges polarized in the direction of the transverse field. One can see in Fig.4(b) that up to some minor discrepancy due to a finite-size effect the two excitation spectra are identical. Moreover, the universal term in the ground-state energy shown in Fig.4(a) is the same for both sets of boundary conditions.

This lead to an important conclusion: $Z_{\text{A-New}} = Z_{\text{Free-ABC}}$. Given that fixed boundary conditions is dual to the free ones, the "New" transverse-polarized boundary conditions have to be dual to the three-state mixed one. In particular, it means that the transverse-polarized boundary conditions are conformally invariant not only for the three-state Potts but also for the four-state Potts models.

Note, that there is no fitting or adjustment parameter in Fig.4(b), thus the agreement between the two towers is spectacular! This made our conclusion on the conformal invariance and on the duality of the transverse-polarized boundary conditions to be solid and independent on any sort of errors associated with an extrapolation. However, to the best of our knowledge, this is the first time the duality of the transverse-polarized boundary conditions is discusses in the context of the four-state Potts model. We therefore would like to present the results that

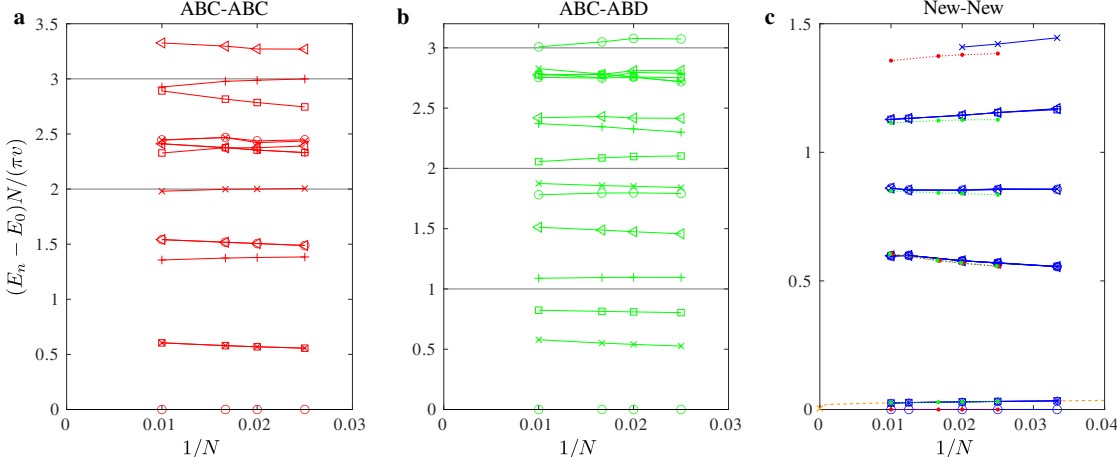

Figure 5: Conformal towers of states of the four-state Potts model with **a** three-state-mixed symmetric and **b** three-state-mixed non-symmetric boundary conditions, and **c** with symmetric boundaries polarized in the direction of the transverse field (following Ref. [24] we use the term "New" boundary conditions). Symbols are DMRG data points extracted from the low-lying energy excitation spectra with velocity $v = \pi/4$. Gray lines are integer levels shown for reference. In **c** the results from **a** (dotted red) and **b** (dotted green) are shown as a reference, the latter is shifted such that it starts at the first excites state of the New-New tower. Each level in New-New tower that matches a level of the ABC-ABD tower is three-fold degenerate. Orange dashed line is a finite-size extrapolation assuming log-corrections in the form derived by Cardy [45].

provide an additional check to our conclusion.

Let us first consider the symmetric New-New boundary conditions. If we assume the duality between the transverse polarized and the three-state mixed boundary conditions, one can expect:

$$Z_{New-New} = Z_{ABC-ABC} + Z_{ABC-ABD} + Z_{ABC-ACD} + Z_{ABC-BCD}. \qquad (12)$$

Because of the symmetry of the model the last three terms are equal $Z_{ABC-ABD} = Z_{ABC-ACD} = Z_{ABC-BCD}$. The spectrum of the ABC-ABC boundary condition is presented in Fig.5(a) and with ABC-ABD boundary conditions - in Fig.5(b). As always, we expect the ground-state of a chain with symmetric edges to belong to the identity conformal tower with $x = 0$. Therefore, we associate the lowest state of the New-New tower with the lowest state of the ABC-ABC tower, as indicated by red dots in Fig.5(c). The first excited state of the New-New tower is therefore associated with the ground-state of the ABC-ABD tower and all higher levels are shown with respect to it. Note that all levels in the New-New tower that match ABC-ABD tower are three fold degenerate, as expected from Eq.12. In particular, the level that matches both ABC-ABC and ABC-ABD tower (at $(E_n - E_0)N/(\pi v) \approx 0.6$) is five-fold degenerate: two-fold degeneracy comes from the degenerate first excitation in ABC-ABC tower and three-fold degeneracy comes from the non-degenerate first excited state in ABC-ABD, ABC-ACD, and ABC-BCD towers.

Finally, let us take another combination of the transverse-polarized and the blob boundary conditions. Again, assuming the duality one can write:

$$Z_{Free-New} = Z_{A-ABC} + Z_{A-ABD} + Z_{A-ACD} + Z_{A-BCD}. \qquad (13)$$

Because of the symmetry of the model the first three terms are identical $Z_{A-ABC} = Z_{A-ABD} = Z_{A-ACD}$. The spectrum of the A-ABC boundary conditions is presented in Fig.6(a) and the spec-

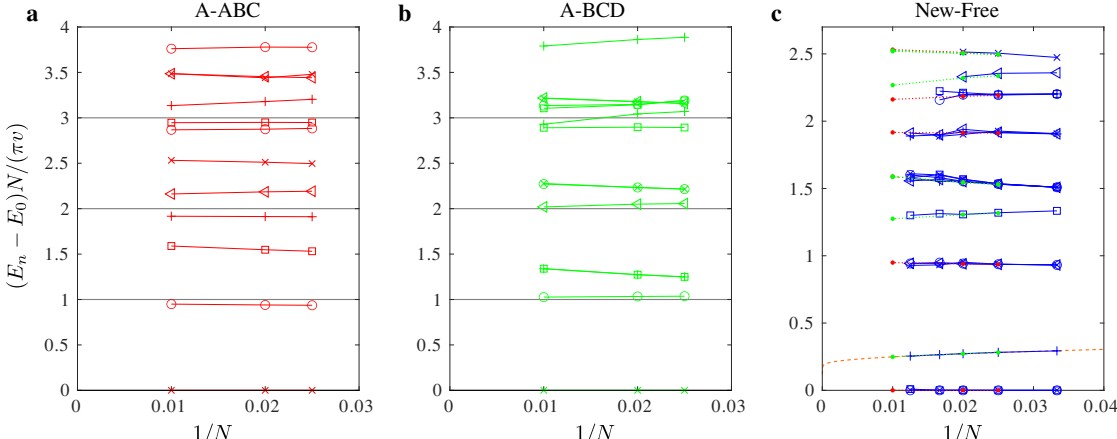

Figure 6: Conformal towers of states of the four-state Potts model with **a-b** fixed boundary condition on one edge and three-state-mixed boundary conditions on the other edge, and with **c** transverse-polarized (New) boundary condition on one edge, while the second edge is free. Symbols are DMRG data points extracted from the low-lying energy excitation spectra with velocity $v = \pi/4$. Gray lines are integer levels shown for reference. In **c** the results from **a** (dotted red) and **b** (dotted green) are shown as a reference, the latter is shifted such that it starts at the first excites state of the New-Free tower. Each level in New-Free tower that matches a level of the A-ABC tower is three-fold degenerate. Orange dashed line is a finite-size extrapolation assuming log-corrections in the form derived by Cardy [45].

trum of the A-BCD ones is shown in Fig.6(b). None of these boundary conditions is symmetric, so we cannot assume the ground-state to belong to the identity tower. Instead, we notice the the ground-state of the New-Free spectrum is three-fold degenerate and therefore we associate the zero-level of the New-Free tower with a zero-level of the A-ABC tower while the first excited state of the New-Free tower we associate with the ground-state of the A-BCD tower. All higher levels of the A-ABC tower (red) and A-BCD tower (green) are plotted in Fig.6(c) with respect to these chosen origin. The numerically obtained degeneracies of the levels always match Eq.13.

# 5 Discussion

In the first part of the paper we have provided numerical evidences that the "new" boundary conditions predicted with boundary conformal field theory by Affleck et al. [24] can be realized in the quantum version of the three-state Potts model by polarizing the edges in the direction of the transverse field. This complements previous DMRG results for conformal towers of states in the quantum three-state Potts model with fixed, mixed and free boundary conditions [16] and completes the numerical realization of all possible conformally-invariant boundary conditions for this model [24,25].

The main conclusion of the paper relies on the empirical observation that the energy spectra of the quantum critical four-state Potts model with A-New and with ABC-Free boundary conditions are identical. This establishes the duality between the transverse-polarized (New) and the three-state-mixed boundary conditions, with one single-particle state suppressed at the edges. Together with the boundary conformal field theory predictions for the three-state Potts model [24] this allow us to say that transverse-polarized boundary conditions are dual to the blob boundary conditions with $Q_1 = Q - 1$, at least for $2 \leq Q \leq 4$. For the transverse-field

Ising model with $Q = 2$ the transverse-polarized boundary conditions are equivalent to the free boundary condition, while suppressing one out of two single-particle state at the edges naturally lead to the fixed boundary conditions; the duality between free and fixed boundary condition is well established. Now, thus far, the duality between the transverse-polarized boundary conditions have been established only for integer values of $Q$. In field theory, however, $Q$ is often treated as a continuous parameter. We hope that our results will stimulate further field theory investigation of the "new" boundary conditions and the duality for an arbitrary values of $Q$.

Furthermore, it would be interesting to check whether the observed duality between the transverse-polarized and the three-state mixed boundary conditions can be generalized to a generic Ashkin-Teller critical model. At this stage we already know that the duality holds at the two special points of this model: at $\lambda = 1$ that corresponds to the symmetric four-state Potts critical point, and at $\lambda = 0$ that corresponds to two decoupled Ising chains. It would be extremely interesting to see whether the duality can be established for a generic Ashkin-Teller model with $0 < \lambda < 1$.

Since three-state mixed boundary conditions of the four-state Potts model are known to be conformally invariant, the established duality implies that transverse polarized boundary conditions are also conformally invariant. This compliment the set of known conformally invariant boundary conditions of the four-state Potts critical theory that up to date was restricted to the blob (free, fixed, mixed) and various loop (periodic, anti-periodic, twisted) boundary conditions. We further check the duality by comparing the energy spectra with New-New and New-Free boundary conditions against the composed dual counterparts. Finally, it would also be interesting to see whether there is a set of boundary conditions dual to the two-state mixed one and what would be the nature of the corresponding boundary term. In the Appendix C we briefly present the results for these boundary conditions.

## Acknowledgments

I would like to thank Frédéric Mila for stimulating questions that lead to investigation reported in this paper. I would also like to thank Frank Verstraete for pointing out that the "new" boundary conditions have been overlooked in our previous numerical study of the three-state Potts model. I am indebted to Ian Affleck for teaching me the basics of boundary conformal field theory in the context of other projects. Numerical simulations have been performed on the Dutch national e-infrastructure with the support of the SURF Cooperative and the facilities of the Scientific IT and Application Support Center of EPFL.

## A  Alternative formulation of the Potts and Ashkin-Teller models

Definition of the three- and four-state Potts models given by Eq.1 are not unique. In this appendix we mention alternative definitions commonly used in the literature.

We start with a $\mathbb{Z}_n$ formulation of quantum three-state Potts model inspired by the corresponding classical Hamiltonian $H = -(J/\beta)\sum_{\langle i,j\rangle}\cos(\theta_i - \theta_j)$, where $\theta_i$ is restricted to the values $0, \pm 2\pi/3$. In the quantum version the Hamiltonian known also as three-state clock model is defined by:

$$H = -\sum_i M_i + M_i^\dagger + R_i^\dagger R_{i+1} + R_i R_{i+1}^\dagger, \tag{14}$$

where

$$M = \begin{pmatrix} 0 & 1 & 0 \\ 0 & 0 & 1 \\ 1 & 0 & 0 \end{pmatrix}; \quad R = \begin{pmatrix} e^{2\pi i/3} & 0 & 0 \\ 0 & e^{4\pi i/3} & 0 \\ 0 & 0 & 1 \end{pmatrix}. \tag{15}$$

This can easily be generalized to the four-state Potts model for which $M$ and $R$ matrices will take the following form:

$$M = \begin{pmatrix} 0 & 1 & 0 & 0 \\ 0 & 0 & 1 & 0 \\ 0 & 0 & 0 & 1 \\ 1 & 0 & 0 & 0 \end{pmatrix}; \quad R = \begin{pmatrix} e^{\pi i/2} & 0 & 0 & 0 \\ 0 & e^{\pi i} & 0 & 0 \\ 0 & 0 & e^{3\pi i/2} & 0 \\ 0 & 0 & 0 & 1 \end{pmatrix}. \tag{16}$$

There are also an alternative formulation of the Ashkin-Teller model where the four-dimensional Hilbert space is defined with the help of two Ising variables. The the Ashkin-Teller model is then defined by the following Hamiltonian:

$$H_{AT} = -h \sum_{j=1}^{N} \left( \sigma_j^x + \tau_j^x + \lambda \sigma_j^x \tau_j^x \right) - J \sum_{j=1}^{N-1} \left( \sigma_j^z \sigma_{j+1}^z + \tau_j^z \tau_{j+1}^z + \lambda \sigma_j^z \tau_j^z \sigma_{j+1}^z \tau_{j+1}^z \right), \tag{17}$$

where $\sigma^{x,z}$ and $\tau^{x,z}$ are Pauli matrices. Similar to the Hamiltonian of the main text given by Eq. the model is critical along $h = J$. At $\lambda = 0$ the model corresponds to two decoupled transverse-field Ising chains. At $\lambda = 1$ the Hamiltonian corresponds to the four-state Potts model.

One can express various boundary conditions in terms of Ising variables by associating A, B, C, and D boundary states from the main text with ↑↑, ↑↓, ↓↑, and ↓↓ of the Ising variables on the edges. For instance, to realize A-A boundary condition, one has to fix Ising variables to ↑↑ state on each edge of the chain. In order to realize ABC-ABD boundary condition, one has to suppress ↓↓ state at the left edge and ↓↑ state at the right edge, keeping the remaining states equally probable.

# B    Characters of the three-state Potts model

Six out of ten primary fields appear in the description of the operators identity $I$ of zero dimension, magnetization $\sigma$ of dimension $1/15$, energy $\epsilon$ of dimension $2/5$, and $\psi$ of dimension $2/3$. The corresponding characters are:

$$\chi_I = \chi_{1,1} + \chi_{4,1}, \qquad \chi_\epsilon = \chi_{2,1} + \chi_{3,1}, \qquad \chi_\sigma = \chi_{\sigma^\dagger} = \chi_{2,3}, \qquad \chi_\psi = \chi_{\psi^\dagger} = \chi_{1,3}. \tag{18}$$

The small-$q$ expansions of the characters for the ten primary fields of the three-state Potts minimal model are given by:

$$\chi_{(1,1)}(q) = q^{-1/30} \left( 1 + q^2 + q^3 + 2q^4 + 2q^5 + 4q^6 + \dots \right), \tag{19}$$

$$\chi_{(2,1)}(q) = q^{-1/30+2/5} \left( 1 + q + q^2 + 2q^3 + 3q^4 + 4q^5 + 6q^6 + \dots \right), \tag{20}$$

$$\chi_{(3,1)}(q) = q^{-1/30+7/5} \left( 1 + q + 2q^2 + 2q^3 + 4q^4 + 5q^5 + 8q^6 + \dots \right), \tag{21}$$

$$\chi_{(4,1)}(q) = q^{-1/30+3} \left( 1 + q + 2q^2 + 3q^3 + 4q^4 + 5q^5 + 8q^6 + \dots \right), \tag{22}$$

$$\chi_{(1,2)}(q) = q^{-1/30+1/8} \left( 1 + q + q^2 + 2q^3 + 3q^4 + 4q^5 + 6q^6 + \dots \right), \tag{23}$$

$$\chi_{(2,2)}(q) = q^{-1/30+1/40}\left(1 + q + 2q^2 + 3q^3 + 4q^4 + 6q^5 + 9q^6 + \dots\right), \tag{24}$$

$$\chi_{(3,2)}(q) = q^{-1/30+21/40}\left(1 + q + 2q^2 + 3q^3 + 5q^4 + 7q^5 + 10q^6 + \dots\right), \tag{25}$$

$$\chi_{(4,2)}(q) = q^{-1/30+13/8}\left(1 + q + 2q^2 + 3q^3 + 4q^4 + 6q^5 + 9q^6 + \dots\right), \tag{26}$$

$$\chi_{(1,3)}(q) = q^{-1/30+2/3}\left(1 + q + 2q^2 + 2q^3 + 4q^4 + 5q^5 + 8q^6 + \dots\right), \tag{27}$$

$$\chi_{(2,3)}(q) = q^{-1/30+1/15}\left(1 + q + 2q^2 + 3q^3 + 5q^4 + 7q^5 + 10q^6 + \dots\right). \tag{28}$$

## C Two-state-mixed boundary conditions

In the main text we presented the results for blob boundary conditions with $Q = 4$ and $Q_1 = 1$ (fixed), $Q_1 = 3$ (three-state mixed) and $Q_1 = Q = 4$ (free) boundary conditions. For completeness let us also present the numerical results for the spectra with $Q_1 = 2$ two-state-mixed boundary conditions. There are three possible combinations. When the pair of components along which we apply the field is the same on both edges, we will call this boundary conditions AB-AB. When only one component coincides, we will refer to these boundary conditions as AB-AC. When the two pairs of components do not overlap we end up with the AB-CD boundary conditions. All other combinations ccan be redused to these three by the symmetry arguments.

Our numerical results for these boundary conditions are presented in Fig.7. The structure of the spectrum excludes the duality between transverse-polarized and two-state mixed boundary conditions. However, if one assumes that every blob boundary condition has the dual one, it would be extremely interesting to understand the nature of the boundary conditions, dual to the two-state mixed one. This question, however, is beyond the scope of this paper.

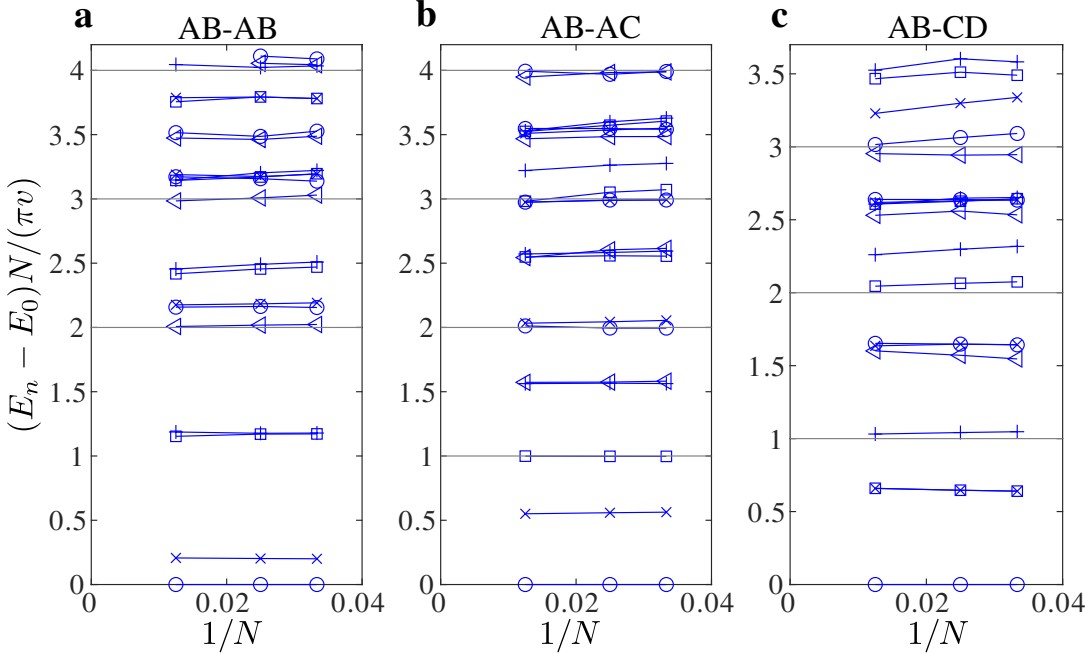

Figure 7: Conformal towers of states of the four-state Potts model with two-state mixed boundary condition. Symbols are DMRG data points extracted from the low-lying energy excitation spectra with velocity $v = \pi/4$. Gray lines are integer levels shown for reference.

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
