# Peer review of "Critical properties of quantum three- and four-state Potts models with boundaries polarized along the transverse field"

_SciPost Physics, doi:SciPost Phys. Core 5, 031 (2022)_

## Round 1 · Referee Report · Anonymous · 2021-10-21

Strengths

1- Extended numerical calculation for non diagonal boundary condition quantum chain

Weaknesses

1- It is seems the author ignore many results in the literature. Due to that he ignores exact results, and report some results (not precise)

Report

The author provide a numerical investigation of the conformal dimensions
happening in some special boundary conditions of the the 3- and 4-state Potts
model. He uses an extension of the Density Matrix Rennormalization Group,
proposed previously by the author previously.

The main result of the paper is the verification that the predicted operator
content predicted by Afleck, Oshikawa and Saleur, in 1998 (the reference
(21) of the paper) is correct.

I have some objections for this paper.
a) Several studies of the operator content of the Potts models were done
in the last 20-25 years, that the author do not compare. Many of the
results were obtained by exploring the fact that the model is a
representation of the Temperley-Lieb algebra ( see for example the papers
of alcaraz, Batchelor,Rittenberg, on the above period).

b) Due to the point (a) several results were derived exploring the equivalence
of energy levels of the the XXZ quantum chain and the Potts (or the
Ashkin-teller). Since the XXZ can be solved by the Bethe ansatz (for some
boundaries), the low-lying levels can be estimated for lattices quantum
chains of quite large chains, or even analytically.

c) Due to the point b) for example, (see eq. 5.11 of Ann.Phys. 182 (1988)
the exact result for the sound velocity
of the 3-state model is \sqrt{3}/2 = 0.866035..., that is different from
the result assumed in the paper (0.857), actually the same problem already
appear in a previous publication of the author (ref.15). In the 4-state
Potts model the author should also know that the exact value is \pi/4=0.78539816
....

d) The point c) wonders me since using only lattices up to size 14, the
authors of J.Phys.A 19,107 (1985), obtain for the sound velocity of the
3-state Potts model a value .858-0.867 (not lattice 100 like the present
paper), this is an indication that the extrapolations may not be considered prop
erly in the present paper.

e) The author also present some considerations about a model (eq. 10) that
he claims to be the Ashkin-Teller model. The quantum Ashkin-Teller has a long
story, and the operator content on several boundaries are already known. The
author would help if write the Ashkin-Teller (also the Potts modes) in terms
of the standard Z(N) operator, satisfying the Z(N) exchange algebra, and
also in terms of two coupled Ising chains. In this formulation will
be more clear what are the mixed boundary conditions.

In summary, my overall vision of the present paper is that the author did
a lot of numerical work in a problem that is already known a lot of results,
without going in the details of the literature, and re deriving proximate
results whose numerically exact results are known.
Due to the above facts I believe this paper will bring confusion in the
literature and I recommend its rejection.

---

## Round 2 · Referee Report · Anonymous (Referee 1) · 2022-3-21

Report

The author made some modifications, and some additions was done that made the paper more clear. It contains the results for the operator content
for the 3- and 4-state models with some kind of open boundary conditions
not studied previously. I recommend the present version of the paper.

---

## Round 2 · Referee Report · Anonymous (Referee 2) · 2022-5-6

Report

In this paper, the author proposes an interpretation for the ‘new’ conformal boundary conditions in the spin-chain formulation of the 3-state and 4-state Potts models. This proposal is supported by careful numerical DMRG work.

This result is new and certainly merits publication in SciPost Physics Core. However, it is not clear to me whether this paper rises to the level of SciPost Physics. Indeed, as already noted, the result is based only on numerics; there is no attempt at a derivation or interpretation in a field theory context. Moreover, an interpretation of the ‘new’ boundary condition has already been known in the transfer-matrix formulation of the 3-state Potts model. While there is a standard correspondence between the spin-chain and transfer-matrix formulation of this model, there is no attempt in this paper to relate the ‘new’ boundary condition in these two formulations.

---

## Round 2 · Author Response

I would like to thank the referee for reading the manuscript and for his/her comments and specially for bringing to my attention a number relevant references. I acknowledge that some numerical results were presented as new while analytical results were available. And I agree with the referee that this would have brought a confusion. This was definitely not intentional and I am very grateful to the referee for pointing this out.

However, all these quantities that I had to evaluate numerically because I did not know that there were exact values were by no means the main goal of the paper, but just a necessary step towards the real goal - the understanding of the new boundary conditions. Let me quote a recent review by Robertson, Jacobsen and Saleur (Journal of High Energy Physics 254, 2019): “The new boundary condition discussed in [Affleck, Oshikawa, Saleur, 1998] is dual to the mixed boundary condition Q = 3, Q1 = 2. It appears however less obvious how to discuss it directly in terms of the original spins, i.e., without invoking duality”. In the present paper I directly address the issue and realize the new boundary condition by applying the local transverse field at the edges. More importantly, I generalize the concept of the new boundary conditions beyond Q=3.

For this reason, I still believe that the main results of the paper – the new transverse-polarized boundary conditions in the four-state Potts model are conformally invariant and dual to the three-state mixed ones – are new and interesting. Therefore, I modified the paper, carefully quoting the exact results and using them as a very useful benchmarks for the method I have used in the paper. I think the revised version of the manuscript should not be confusing any more. By contrast, it brings a very intuitive way to realize new boundary conditions in quantum 1D models.

I hope this will motivate the referee to take another careful look at the revised version of the paper.

Below I provide a point-to-point reply to the concerns raised by the referee:

>> The main result of the paper is the verification that the predicted operator content predicted by Afleck, Oshikawa and Saleur, in 1998 (the reference (21) of the paper) is correct.

The main results of the paper is the realization of the new conformally invariant boundary conditions dual to the three-state mixed ones in the quantum version of the 4-state Potts model by applying the transverse field at the boundary. The operator content of the 3-state Potts model has been verified by the present author in Ref.[15] and by our colleagues in Ref.[19]. In Ref.24 (former Ref.21) the authors provide, as they call it, an indication that new boundary conditions can be obtained by reversing the sign of the transverse field at the boundary. However, to the best of my knowledge, this indication has never been verified numerically. And, although this verification was by no means the main goal of the paper, I decided to include it into the manuscript for completeness.

>>a) Several studies of the operator content of the Potts models were done in the last 20-25 years, that the author do not compare. Many of the results were obtained by exploring the fact that the model is a representation of the Temperley-Lieb algebra ( see for example the papers of alcaraz, Batchelor,Rittenberg, on the above period).

I thank to the referee for pointing out these references. I included the exact results available for the models and the corresponding references into the revised version of the manuscript.

>> b) Due to the point (a) several results were derived exploring the equivalence of energy levels of the the XXZ quantum chain and the Potts (or the Ashkin-teller). Since the XXZ can be solved by the Bethe ansatz (for some boundaries), the low-lying levels can be estimated for lattices quantum chains of quite large chains, or even analytically.

Bethe anzatz allows to obtain the spectrum for very limited number of boundary conditions (as also pointed by the referee). Among seven different boundary conditions considered in Ann.Phys. 182 (1988), only one is defined on an open chain – the one with free boundary conditions; the rest are defined on a loop (periodic, anti-periodic, twisted etc). To reach the main goal of the paper – understanding of the new boundary conditions, it is crucial to get access to other types of boundary conditions. To the best of my knowledge, neither non-symmetric three-state mixed nor transverse-polarized boundary conditions have even been solved by mapping to the XXZ model and using the Bethe ansatz.

>>c) Due to the point b) for example, (see eq. 5.11 of Ann.Phys. 182 (1988) the exact result for the sound velocity of the 3-state model is \sqrt{3}/2 = 0.866035..., that is different from the result assumed in the paper (0.857), actually the same problem already appear in a previous publication of the author (ref.15).

I would like to thank the referee for pointing to me this exact result. I was not aware of it. In the revised version of the paper, I have replaced the value of the sound velocity for the 3-state Potts point with the exact value. However, I would like to draw the referee’s attention that the agreement between these two values is within 1%.

>>In the 4-state Potts model the author should also know that the exact value is \pi/4=0.78539816

I specially thank the referee for pointing out this exact results and I am very glad to see that my numerical estimate of the sound velocity v~0.785 is in excellent agreement with it. I replaced the sound velocity with its exact value throughout the paper, and use the numerical result for the velocity as a benchmark.

>>d) The point c) wonders me since using only lattices up to size 14, the authors of J.Phys.A 19,107 (1985), obtain for the sound velocity of the 3-state Potts model a value .858-0.867 (not lattice 100 like the present
paper), this is an indication that the extrapolations may not be considered properly in the present paper.

In the aforementioned reference, the exact diagonalization (Lanczos) has been used. As explained in Ref.[19] for small systems like N=14 the DMRG method used in the present manuscript is rigorously equivalent to exact diagonalization. And once more I would like to bring to the referee’s attention that the discrepancy is about 1%.

>>e) The author also present some considerations about a model (eq. 10) that he claims to be the Ashkin-Teller model. The quantum Ashkin-Teller has a long story, and the operator content on several boundaries are already known. The author would help if write the Ashkin-Teller (also the Potts modes) in terms of the standard Z(N) operator, satisfying the Z(N) exchange algebra, and also in terms of two coupled Ising chains. In this formulation will be more clear what are the mixed boundary conditions.

Following the referee’s suggestion I provide an appendix with alternative formulations of the Potts and the Ashkin-Teller models. In my opinion, mixed boundary conditions in terms of Ising variables are much less intuitive. For instance, the ABD boundary state in this formulation corresponds to a couple of Ising variables where the state down-up is fully suppressed, while up-up, up-down and down-down are realized with equal probabilities. Such state cannot be realized with any local fields acting on Ising variables. But, I guess, it is a matter of taste.

---

## Editorial Decision

published